# Association of Physical Educators’ Socialization Experiences and Confidence with Respect to Comprehensive School Physical Activity Program Implementation

**DOI:** 10.3390/ijerph191912005

**Published:** 2022-09-22

**Authors:** Christopher Barton Merica, Cate A. Egan, Collin A. Webster, Diana Mindrila, Grace Goc Karp, David R. Paul, Karie Lee Orendorff

**Affiliations:** 1School of Health and Applied Human Sciences, University of North Carolina Wilmington, Campus Box 5956, Wilmington, NC 28401, USA; 2Department of Movement Sciences, University of Idaho, 875 Perimeter Drive MS 2401, Moscow, ID 83844, USA; 3School of Sport, Exercise and Rehabilitation Sciences, University of Birmingham Dubai, Dubai P.O. Box 341799, United Arab Emirates; 4Department of Leadership, Research, and School Improvement, University of West Georgia, 1601 Maple Street, Carrollton, GA 75006, USA; 5Department of Health and Human Development, Montana State University, Culbertson Hall 100, Bozeman, MT 59717, USA

**Keywords:** whole of school physical activity, physical education, physical education teacher education, teacher socialization, role breadth self-efficacy

## Abstract

Comprehensive school physical activity programs (CSPAPs) are recommended to support physical education (PE) and increase the amount of physical activity (PA) youth receive each day. However, adoption of CSPAPs in the United States is low. PE teachers are well positioned to lead the implementation of CSPAPs, but research is needed to better understand (a) PE teachers’ confidence to assume the multiple roles involved with CSPAP implementation and (b) the factors that are associated with such confidence. This study examined PE teachers’ role breadth self-efficacy (RBSE) as a measure of PE teachers’ CSPAP-related confidence and its association with seminal life experiences as framed within teacher socialization theory. A survey was emailed to a stratified-random sample of 2976 PE teachers and distributed on social media, garnering a total of 259 responses. Exploratory structural equation modeling supported a three-factor solution for teacher socialization variables (acculturation, professional socialization and organizational socialization), in line with the theoretical framework, and a single factor solution for RBSE. Professional socialization and organizational socialization were significant predictors of RBSE, and qualitative data from open-ended survey questions supported these relationships. The results highlight the importance of preservice teacher education and current employment contexts in PE teachers’ CSPAP-related confidence.

## 1. Introduction

Obesity and associated health problems are widely prevalent among school-age youth internationally [1]. Participating in regular physical activity (PA) reduces adverse health effects related to sedentarism, but 81% of adolescents do not meet PA guidelines [2]. The school setting is identified as a key intervention point for increasing PA behaviors of school-aged youth due to: (a) the prominent amount of time youth spend at school and (b) the number of children and adolescents who attend school regularly (95% of young people [3,4]). To combat youth inactivity, targeted “whole-of-school” PA programs are recommended by national and international organizations [1,3,4].

In the United States (U.S.), the comprehensive school PA program (CSPAP) has emerged as a whole-of-school approach to PA promotion with two major goals: (a) equip youth with the knowledge, skills, and confidence to engage in a lifetime of participation in PA, and (b) ensure youth meet the recommendation of at least 60 min of moderate-to-vigorous PA each day [4]. A CSPAP is a five-component framework that includes: (a) quality physical education (PE), (b) PA during school, (c) PA before and after school, (d) staff involvement, and (e) family and community engagement [4]. PE is an essential foundation of a CSPAP, as it uniquely foregrounds teaching youth knowledge and skills for a physically active lifestyle, while other PA opportunities before, during and after school can be used as needed to support PE and help to ensure youth meet PA guidelines [5]. The staff involvement and family and community engagement components of a CSPAP serve as the support system for program implementation [5]. Recently, the Centers for Disease Control and Prevention (CDC) adopted the CSPAP framework as the national framework for school PE and PA in the U.S. [6].

PE teachers are called upon to be school PA leaders (PALs), who organize, lead, and promote PA through the implementation of a CSPAP [7,8,9]. The involvement of PE teachers in school wide PA promotion thus entails expanded professional roles beyond teaching PE [10], and is tied to an extensive range of recommended knowledge (e.g., behavior change theories, benefits of PA, trends and issues related to PA) and skills (e.g., advocating for school-based PA, promoting outside-of-school PA, evaluating school-based PA programming [11]. However, based on research conducted at a national scale, there appears to be a substantial number of PE teachers who work at schools without a CSPAP and whose leadership and involvement in CSPAP adoption may be minimal [5]. Therefore, it is imperative to further investigate PE teachers’ involvement with CSPAPs, as such research will enable future interventions and teacher professional development to better support PE teachers in helping to advance CSPAP implementation efforts [12].

## 2. Role Breadth Self-Efficacy

One perspective to examine PE teachers’ involvement with CSPAPs, that has not previously been explored, is that of teacher confidence. Theoretically, teacher confidence closely aligns with the concept of self-efficacy, which refers to people’s judgements about their capability to perform particular tasks [13], such as taking on a particular role or occupation [13,14]. Self-efficacy has received considerable investigative attention in the field of education [13,14,15,16], and more specifically in PE [17,18,19,20,21]. Teachers who possess high levels of self-efficacy demonstrate effective teaching behaviors in the classroom [22,23]. For example, PE teachers with high levels of self-efficacy were more skillful in using teaching strategies and classroom management strategies [20,21,24], and were more likely accept new ideas and roles within the school setting [18,25,26]. Additionally, other educational research found that teachers with high self-efficacy exerted more effort to overcome problems they encountered [13,14,18,27]. Overall, self-efficacy thus constitutes an important motivational construct that influences a teacher’s goals and choices [15,28,29].

Given that CSPAP leadership and involvement entail multiple roles for PE teachers, a particularly relevant type of self-efficacy for the present study is role breadth self-efficacy (RBSE), which applies to situations where professionals need to extend their scope of work to encompass a broader range of skills and responsibilities than what might traditionally have been expected. RBSE is “the extent to which people feel confident that they can carry out a broader and more proactive role, beyond traditional prescribed technical requirements” [30] (p. 835). Specifically, RBSE has been conceptualized as confidence to carry out proactive, interpersonal, and integrative professional roles [30,31]. All three of these roles are clearly apparent in the work envisioned for implementing a CSPAP. As there is limited accountability for CSPAP implementation in the U.S. [32], PE teachers would likely need to be proactive and self-directed in taking on the role of being a PAL to initiate and grow school wide PA promotion. Additionally, CSPAP leadership and involvement would entail a strong interpersonal component [8]. PE teachers serving as PALs would need to motivate and bridge the PA promotion efforts of other school staff, families, and community partners. Whole-of-school PA programming would also involve integrative work for PE teachers, as they would need to find ways to both support the educational goals of PE and increase students’ daily PA through multiple program components [5].

## 3. Teacher Socialization

A relevant lens for examining and understanding PE teachers’ confidence (i.e., RBSE) to lead and be involved with CSPAP is teacher socialization theory. Socialization theory investigates associations between the lived experiences and current teaching behaviors, beliefs, and dispositions of PE teachers [33,34]. Specifically, it considers lived experiences over a three-phase, non-linear process of a teacher’s lifetime [35]. These phases include: (a) acculturation (i.e., positive or negative experiences in childhood as a K-12 student, which lead to the development of beliefs and attitudes toward the teaching profession and PE), (b) professional socialization (i.e., socialization into PE as preservice teachers in PE teacher education [PETE] programs), and (c) organizational socialization (i.e., socialization into the role as a teacher influenced by school contexts [33,34,35,36].

Most of the socialization literature has focused on each phase of socialization separately [36], with minimal investigation into the role of teachers’ involvement with school-based PA promotion [35]. The acculturation literature suggests that in-service PE teachers are highly influenced by their K-12 PE teachers and sport coaches [37,38], although the association of PE teachers’ acculturation experiences and CSPAP-related confidence is unknown [7,35,36]. Regarding professional socialization, there is an abundance of recommendations for PETE programs to support teacher candidates’ development of knowledge and skills to serve as PALs and be involved with CSPAPs [9,11], and a burgeoning line of empirical study suggests that PETE programs can provide positive CSPAP-related learning experiences for preservice PE teachers [39,40,41,42,43]. However, minimal research has investigated the association of professional socialization and in-service PE teachers’ confidence to be involved with CSPAPs [5,7]. Finally, research related to organizational socialization of PE teachers indicates that perceived school support is an important factor in CSPAP involvement [44], but little is known about which aspects of organizational socialization (i.e., administrative support, co-teacher beliefs, school/district policies, availability and/or condition of building facilities, equipment/materials, budget) are linked to PE teachers’ RBSE in regard to CSPAP implementation [5,35,44,45].

## 4. Purpose of the Study

Documenting PE teachers’ RBSE with respect to CSPAP leadership and involvement, as well as investigating the role of socialization in teachers’ CSPAP-related RBSE, are important steps in advancing theory and research that identify and explain teacher-level factors influencing the implementation of schoolwide PA promotion initiatives. Such scholarship may provide evidence to inform the work of interventionists, teacher educators, and others who provide both preservice and continuing professional development for PE teachers. For instance, if certain phases of socialization (e.g., professional socialization, organizational socialization) are determined to be significant factors in PE teachers’ CSPAP-related RBSE, then education during specific phases of teachers’ career development could include targeted learning experiences designed to build PE teachers’ confidence in line with RBSE. In turn, enhancing PE teachers’ CSPAP-related RBSE could lead to an increase in the rate of CSPAP adoption. Thus, the purpose of this study was to examine the association of PE teachers’ CSPAP-related RBSE and socialization (acculturation, professional socialization, and organizational socialization). The specific aims were to: (a) describe PE teachers’ RBSE and socialization experiences through a descriptive analyses and qualitative analyses, (b) identify latent socialization factors, and (c) examine the association of RBSE and socialization factors.

## 5. Methods

### 5.1. Participants

A total of 259 PE teachers participated in this study. The sample had a balanced gender distribution (50% female, 47.1% male, 1.8% transgender, 1.2% preferred not to say). Participants’ age ranged between 20–24 (3.6%), 25 and 34 (32.7%), 35 and 44 (28%), 45 and 54 (22%), 55 and 64 (11.3%), and 65 and older (2.4%). Most of the respondents (79.4%) identified as White, 7.6% identified as African American or Black, 3.5% identified as Asian, 3.5% identified as more than one race, 3.5% preferred not to disclose their race, 1.8% identified as American Indian, and 0.6% identified as Alaska Native. Approximately 14.5% of the sample indicated having a Hispanic descent. Most participants received their teacher certification training to become a PE teacher from a university/college PETE program (87.2%). Approximately 6.7% did not receive formal training to become a PE teacher, and 6.2% had completed an alternative licensure (e.g., online certification program), and 34.2% were National Board certified.

### 5.2. Instrumentation

A survey instrument, developed and validated in a previous research study [46], was used to collect data for the present investigation. The instrument measures (a) CSPAP-related socialization experiences in each socialization phase, (b) RBSE to be a PAL and implement a CSPAP, and (c) background/demographic variables. The survey consists of five sections, which are preceded by an introduction and informed consent that include a stated purpose and survey directions. The survey introduction and each subsequent section provide participants with an overview of the CSPAP framework and include the definition of a CSPAP based on previous literature [5,44,47]. At the end of the survey, participants were given the opportunity to submit their name and email address to be entered in a drawing to win a $50 Amazon gift card and/or be interviewed by members of the research team to further discuss their CSPAP perceptions and experiences for future research endeavors.

### 5.3. Procedures

All research activities were approved by the first author’s university Institutional Review Board prior to the initiation of this study. The population of interest was K-12 public school in-service PE teachers in the U.S. Stratified random sampling was used to obtain a proportionately random national sample of schools from which to identify PE teachers [48]. Stratified random sampling has numerous applications and benefits, such as studying population demographics, and is considered a precise metric to represent a larger population [48]. The strata for our sample of PE teachers were characterized by the state in the U.S. where PE teachers taught and the grade level (elementary, middle, or high school).

To limit instances of over-representation of PE teachers from the same school district or school, and develop a system of random selection, sampling procedure rules were implemented. First, contact information for every school district in each state was gathered from the National Center for Education Statistics. Then, the proc survey select procedure (SAS 9.4; Cary, NC, USA) was used to randomly select 20 schools from each grade level in each state. Contact information of potential PE teachers for the study was identified from the school district website of the selected schools. A maximum of two PE teachers from each school level could be chosen. Furthermore, researchers set a goal of selecting only one PE teacher per school. In all, 60 PE teachers (i.e., 20 elementary, 20 middle school, and 20 high school) were selected from each state (totaling 3000 teachers). Sample size goals were set based upon previous nationwide survey research of school professionals and CSPAP involvement [44,47,49,50]. The research team gathered and organized the teachers’ names and email addresses, along with their identifying states, school districts, school names, and education levels using an Excel spreadsheet.

There were several sampling challenges. At the school level, teachers’ email addresses sometimes were unavailable on school websites. In these cases, the sampling rule was extended to no more than two PE teachers chosen from a given school. Additionally, the number of school districts for each state varied greatly. For example, Hawaii operates their entire school system under one school district, while Alaska has a limited number of school districts outside of their major metropolitan areas (i.e., Anchorage, Fairbanks, Juneau). Due to these issues, as a last resort researchers selected PE teachers from school districts and individual schools with the most available contact information if the aforementioned sampling procedure rules could not be met.

Once the target sample size was met (i.e., 3000 teachers), a blanket email was sent to all email addresses identified for the PE teachers inviting them to participate in the study. In the email, teachers were told of the purpose of the study, that completing the survey would enter their name into a drawing for a $50 Amazon gift card, and to use an embedded URL link to complete the survey in Qualtrics. A total of 2976 emails were successfully delivered (24 inactive emails). A five-week window was provided for participants to complete the survey. Follow-up invitation emails to participate in the study were sent to non-responders four consecutive weeks after initial contact.

After initial contact and follow-up email reminders, a total of N = 199 PE teachers had responded to the survey (7% response rate). Due to a low response rate from the stratified sample, researchers decided to distribute the survey link via social media (i.e., Facebook) to a PE-based group (i.e., Health and Physical Education Teaching Resources), which is followed by in-service PE teachers across the U.S. The survey was posted twice on social media within 21 days, generating an additional N = 60 responses (N = 259). One-way ANOVA results showed that item responses from the stratified random sample did not vary significantly from those collected via social media (F(1257) statistics ranged between 0.12, *p* = 0.914 and 3.509, *p* = 0.062).

### 5.4. Data Analysis

For the purposes of this study, 55 items from the survey, which originally included a total of 99 items, were selected for analysis due to sample size requirements. To select these items, four members of the research team (i.e., first, second, third, and fourth authors) conducted a thorough content analysis to determine which items best represented their corresponding construct and addressed the intended content related to PE teachers’ CSPAP-related RBSE. Additionally, items were selected based upon representation and performance measures related to underlying theoretical/conceptual (i.e., socialization phases, RBSE) and statistical criteria (i.e., item factor loadings, model fit indices). The final set of 55 items included 31 items to measure the three socialization phases (6 items for acculturation, 12 items for professional socialization, and 13 items for organizational socialization); 3 items to measure RBSE; 4 open-ended questions to provide complementary qualitative data intended to enrich and contextualize participants’ responses to Likert-type items on the survey [51]; and 17 demographic questions.

#### 5.4.1. Descriptive Analysis

The first step in analyzing data was examining the distribution of survey responses. Descriptive statistics were calculated (i.e., mean [M] and standard deviation [SD]) to determine the extent to which respondents endorsed each survey item. Univariate skewness and kurtosis coefficients and Mardia’s multivariate indices of skewness and kurtosis [52] were then calculated to further examine the distribution of survey responses and determine whether the data met the assumption of multivariate normality. Univariate skewness coefficients larger than two and univariate kurtosis coefficients larger than seven indicate a non-normal distribution [53,54]. Although there are no generally accepted guidelines regarding the values of univariate kurtosis that indicate multivariate non-normality, the research literature suggests that data with multivariate kurtosis larger than three may produce biased results with maximum likelihood (ML) estimation [55,56].

#### 5.4.2. Exploratory Structural Equation Modeling (ESEM)

An exploratory factor analysis (EFA) within the ESEM framework was used to identify the factors underlying the data and examine structural relationships [53]. While traditional structural equation modeling (SEM) relies on a confirmatory factor analysis (CFA), ESEM estimates an exploratory measurement model with rotations and yields a more realistic representation of the data by allowing items to cross-load [54], and the exploratory approach helps avoid item misspecification [53]. Research using simulated data has shown that taking cross-loadings into account increases estimation precision, whereas, fixing even very small cross-loadings such as 0.100 to zero, may induce significant estimation inflation and bias in parameter estimates [53,57]. In addition to EFA, ESEM allows the specification of covariates and structural coefficients, and calculates goodness of fit indices [58,59]. Therefore, ESEM estimates the EFA model, while including the methodological advances of a CFA and SEM, by assessing model fit and allowing the estimation of structural coefficients [58,59]. A total of 31 survey variables were used to examine the socialization factors (i.e., acculturation professional socialization, and occupational socialization [35]) underlying the data, and three variables to separately estimate a single factor measuring RBSE. With a sample size of 259, the cases per variable ratio exceeded the recommendations of 2 to 5 cases per variable or at least 100 subjects [60,61] and 20 subjects per factor [62]. The Mplus 8 statistical software was used to conduct latent variable modeling procedures. Survey responses were standardized and used as observed indicators. The estimation method was maximum likelihood (ML) with Geomin rotation. This procedure provides the most accurate results when variables are continuous, and data meet the assumption of multivariate normality [56].

Solutions with differing numbers of socialization factors were examined. Further, the relationship between socialization factors and RBSE factor scores by specifying RBSE as a dependent variable in the ESEM model was estimated. The optimal solution was selected based on the interpretability of the factors and theoretical criteria. Specifically, statistical criteria consisting of the number of eigenvalues larger than one, the examination of the scree plot, and the following goodness of fit indices were identified: (a) chi-square (χ^2^) and its *p*-value, (b) χ^2^ divided by degrees of freedom (χ^2^/df), (c) Tucker-Lewis index (TLI), (d) comparative fit index (CFI), and (e) standardized root mean square residual (SRMR), and (f) root mean square error of approximation (RMSEA) and its 90% confidence interval (CI).

The χ^2^ test measures overall model fit. Non-significant χ^2^ values show good fit to the data [63]; however, larger models and non-normal data often inflate the χ^2^ coefficient. This limitation was addressed by using χ^2^/df to assess model fit. When χ^2^/df < 3 the model is determined to have a good fit to the data [56]. The TLI and CFI values larger than 0.95 indicate excellent fit, values larger than 0.90 show good fit, whereas values lower than 0.90 indicate poor model fit [64]. RMSEA and SRMR values smaller than 0.05 are evidence of excellent fit, values ranging between 0.05 and 0.08 show good fit, values ranging between 0.08 and 0.10 indicate only acceptable fit, whereas values above 0.10 show poor fit [64].

#### 5.4.3. Qualitative Data Analysis

The four open-ended survey questions were analyzed to better understand the participants’ perceptions and experiences as they relate to each factor (i.e., acculturation, professional socialization, organizational socialization, and RBSE). In total, 57 participants responded to the open-ended acculturation question, 55 for the professional socialization question, 51 for the organizational socialization question, and 49 for the RBSE question. Open-ended questions within surveys provide an opportunity to capture information that cannot be easily captured in closed-ended questions and further explain quantitative results [51]. Two members of the research team (i.e., first and second authors) coded the data and conducted thematic analysis [65] by looking for initial codes and categories. Overall, the open-ended responses ranged from short statements to longer narratives, all of which were captured for data analysis. However, due to limited data the researchers specifically looked for emerging salient points across categories as opposed to developing themes [66]. Open-ended survey response data, such as salient points, are used to enhance, confirm, and/or refine the story told through quantitative data [51,66]. Researchers assigned pseudonyms throughout this article to protect respondents’ anonymity.

## 6. Results

### 6.1. Descriptive Analyses

For detailed information about the participant’s employment region, experience level, and prior CSPAP knowledge see Table 1. As indicated in Table 2, most participants reported receiving a high-quality PE experience as a K-12 student. Among the items designed to measure acculturation, the item with the highest ratings was “As a K-12 student, at least one of my physical education teachers implemented a physical education program that included: standards-based instruction, assessment of student learning, opportunities to learn, opportunities for moderate-to-vigorous physical activity” (M = 4.56, SD = 1.428). Additionally, most respondents highly rated their PETE-based training to teach PE. In the group of items intended to measure professional socialization, the item with the highest ratings was “My teacher certification program prepared me to develop a physical education program that includes standards-based instruction, assessment of student learning, opportunities to learn, opportunities for moderate-to-vigorous physical activity” (M = 5.09, SD = 0.877). Regarding organizational socialization, participants perceived their current school facilities and resources to be important for their CSPAP involvement. The item with the highest ratings was “Indoor and outdoor physical activity facilities/resources (e.g., gym space, weight room, outdoor green space) positively influence my CSPAP involvement” (M = 4.36, SD = 1.058). Lastly, participants felt confident to be leaders of PA in their school and implement a quality physical education program. The two items with the highest ratings measuring RBSE were “I feel confident implementing physical education program that includes standards-based instruction, assessment of student learning, opportunities for moderate-to-vigorous physical activity” (M = 5.26, SD = 0.958) and “I feel confident being a physical activity leader for my school(s) (e.g., organize physical activity opportunities for students outside the classroom, promote physical activity to staff and families/community” (M = 4.94, SD = 1.030).

The survey variables used in this study had an approximately normal distribution. Indices of univariate skewness ranged between −1.843 and −0.782 while indices of univariate kurtosis ranged between 0.46 and 5.35 (Table 2). Mardia’s coefficients of multivariate skewness and kurtosis were 1.601 (*p* = 0.112) and 2.321 (*p* = 0.092), respectively. These indices showed that survey responses had a univariate and multivariate normal distribution. The proportion of missing values ranged between 1% and 12% per survey item. Little’s MCAR test showed that their distribution was completely random (χ^2^ = 638.062, DF = 701, Sig. = 0.957). Therefore, to avoid losing data, missing values were imputed using the expectation-maximization algorithm.

### 6.2. Socialization Factors


Exploratory procedures yielded three eigenvalues larger than one and the scree plot indicated that 3 or 4 factors may underlie the data. Therefore, models with three and four factors were estimated and compared. As indicated in Table 3, the four-factor solution (Model 1) had a slightly better fit to the data but included several cross-loading items and the factors did not have strong theoretical support. The 3-factor solution (Model 2) included only two cross-loading items and the factors clearly described distinct dimensions of socialization (i.e., acculturation, professional socialization, and occupational socialization [35]). Accordingly, the three-factor solution was selected as optimal for our data. The two cross-loading items were sequentially removed, and a simple structure was obtained. The two items removed were: (1) “As a K-12 student, at least one of my physical education teachers organized physical activity opportunities for my family/community (e.g., 5K events, family fitness nights)”, and (2) “Most teachers at my school provide activity breaks in the classroom, as a break, or as part of academic work”. Removing the cross-loading items significantly improved the model fit (Table 3). The final factor structure included only items with loadings above 0.320, no free-standing items, cross-loading items, or items with non-significant loadings (α = 0.05 [67]). Specifying RBSE factor scores as a dependent variable of the three identified factors further improved model fit. As indicated in Table 3, the structural model (Model 4) had a very good fit to the data.

Table 4 lists the items included in each factor, and reports the factor loadings, standard errors, t statistics, *p* values. The same table includes the ESEM path coefficients, factor covariances and factor correlations. In addition, open-ended questions accompanied by compelling quotes to augment quantitative data are included within their corresponding factor. Further information about each factor is provided below.

#### 6.2.1. Acculturation

The first socialization factor included five items measuring acculturation. Loadings on this factor ranged between 0.405 and 0.722. The item with the highest loading was “As a K-12 student, at least one of my physical education teachers was considered the physical activity leader for the school (e.g., organized physical activity opportunities for students outside the classroom, promoted physical activity to students”; 0.722). In addition, the second highest loading item was “As a K-12 student, at least one of my physical education teachers implemented a physical education program that included: standards-based instruction, assessment of student learning, opportunities to learn, opportunities for moderate-to-vigorous physical activity”; 0.538). Cronbach’s coefficient α was 0.728 for the acculturation factor.

Participants’ open-ended responses regarding acculturation revealed that most respondents did not have many opportunities to experience CSPAP-related activities beyond PE, or were limited to special events (e.g., school fun run, field day). Due to the lack of opportunities available as a K-12 student, PE teachers were motivated to lead these types of programs as an in-service teacher. Cindy wrote, “There were very few opportunities [in] my school district. I hope to change that at my school”. Although respondents expressed there were not many CSPAP-related opportunities as a K-12 student, many teachers conveyed that their PE teacher served as a PAL (i.e., taught quality PE, promoted PA beyond PE). “I had many opportunities to engage in vigorous activity and learn and practice motor skills in physical education. Activity outside of school was encouraged and promoted” (Rodney). These results support the quantitative results as highest item factor loadings revolved around participants receiving a strong PE program and having K-12 PE teachers who enacted behaviors consistent with those recommended for a PAL. However, items focused on expanded PA opportunities available for students, staff/faculty, and community members had lower item factor scores (see Table 4).

#### 6.2.2. Professional Socialization

The second socialization factor included twelve items measuring professional socialization. Items in this factor had loadings between 0.515 and 0.887. The two items with the highest loadings were “My teacher certification program prepared me to develop physical activity initiatives for school staff/faculty (e.g., fitness programs/events for teachers, health screening for teachers, staff training for physical activity promotion)”; (0.887) and “My teacher certification program prepared me to evaluate current physical activity offerings in K-12 school environments (e.g., before/after school, during school, facilities, equipment resources); (0.885). Cronbach’s coefficient α for the PS factor was 0.955.

The open-ended question responses with respect to professional socialization supported the factor loadings. Participants discussed their training experiences for expanded PA opportunities before or after school. Participants also mentioned experiences with the staff involvement component of a CSPAP within practicum coursework or student teaching. As Ted explained, “My coordinating teacher and I led several after school fitness classes for staff”. The degree of participants’ CSPAP training experiences for was mixed. For instance, some participants shared that their program did not specifically train for CSPAP implementation, however their training had a major focus on schoolwide PA promotion. As Peter wrote, “CSPAP wasn’t a thing when I went to school [PETE], but schoolwide PA promotion was…finding different avenues to promote PA to the community through newsletters, events, posters”. In addition, based upon professional socialization experiences, many participants felt prepared to implement a CSPAP. “My teacher certification program has their stuff together…I am so much better prepared and educated to implement these programs [CSPAP]” (Violet).

#### 6.2.3. Organizational Socialization

The third socialization factor included twelve items measuring organizational socialization. Items in this factor had loadings between 0.362 and 0.872. The two items with the highest loadings were “Teachers/faculty expect me to implement CSPAP” (0.872) and “Teachers/faculty positively influence my current CSPAP involvement” (0.815). Cronbach’s α was 0.907 for the organizational socialization factor. Open-ended responses from participants support and expand upon this factor. Participants acknowledged they feel motivated to lead and be involved with CSPAP implementation because they have support from their students and other staff. For example, Ted wrote, “Students love participating in PA which influences my desire to organize these events [CSPAP]”. In addition, low factor scores for the item, “Indoor and outdoor physical activity facilities/resources (e.g., gym space, weight room, outdoor green space) positively influence my CSPAP involvement” (0.404) was affirmed by qualitative responses. For example, many participants felt a lack of resources, such as support from administrators and other school staff, and/or funding, negatively impacted their CSPAP involvement and implementation success. “The environment at my school is negative, and I have found that when I do events after school for families, they are not well attended” (Jon).

#### 6.2.4. RBSE Factor

The three items included in the RBSE factor had loadings between 0.756 and 0.920. The item with the highest loading was “I feel confident implementing multiple components of CSPAP (e.g., before/after school physical activity, staff involvement” 0.920). The item with the second highest loading was “I feel confident implementing physical education program that includes standards-based instruction, assessment of student learning, opportunities for moderate-to-vigorous physical activity” (0.870). Cronbach’s α for the RBSE factor was 0.835.

Open-ended data converged with these strong RBSE item factor loadings, as participants noted they felt overwhelmingly capable and confident to implement CSPAP in their schools. However, participants also expressed reluctance to implement CSPAPs because of barriers related to time, support from school peers, or compensation. Brian stated, “Confidence and time to do it are two different things…I won’t work for free. The before and after school programming takes time from my family time. It’s not worth it to me”. Additional evidence of reluctance included statements about feelings of isolation to organize and lead a CSPAP and/or not having support from school administers and faculty.

#### 6.2.5. Relationships between Socialization Factors and RBSE

The ESEM results showed that the professional socialization (estimate = 0.246, *p* < 0.001) and organizational socialization (estimate = 0.353, *p* < 0.001) factors were significant predictors of RBSE factor scores, whereas the acculturation factor was not (estimate = −0.005, *p* = 0.935). As indicated in Table 4, the organizational socialization—professional socialization covariance was statistically significant (estimate = 0.470, *p* < 0.001), whereas the organizational socialization—acculturation (estimate = 0.142, *p* = 0.190) and PS-AC (estimate = 0.119, *p* = 0.298) covariances were not.

The open-ended survey responses support professional socialization and organizational socialization as predictors of PE teachers’ RBSE. For example, participants noted their PETE training experiences for CSPAP implementation afforded them the confidence to implement CSPAP as an in-service teacher. Scott explained, “In my [preservice] field experiences I implemented components of CSPAP in a real school setting, which prepared me to implement CSPAP”. Participants also discussed the important influence their current school environment (i.e., policy, support, resources) has on CSPAP implementation. Jon emphasized this point, indicating “Administrative support and students positively influence my CSPAP facilitation and implementation”. Furthermore, open-ended responses confirmed the relatively less important influence of acculturation on RBSE. For example, participants expressed a lack of CSPAP opportunities and participation as a K-12 student, or they did not remember their acculturation experiences. Violet explains, “I don’t have any recollection of physical education…physical education programs were in place, but CSPAP was not established in my K-12 schools”.

## 7. Discussion

The purpose of this study was to examine the association of PE teachers’ CSPAP-related socialization experiences and RBSE. This study adds to previous research on PE teachers’ perceptions related to CSPAP adoption [44,47] and builds on the theoretical basis for understanding PE teachers’ CSPAP involvement.

**Psychometric analysis.** Psychometric analysis of the survey instrument identified a three-factor solution for the socialization items consistent with the three established phases of teacher socialization in the literature. In addition, a single-factor solution framed around RBSE was found for the three items measuring confidence to be a PAL and implementor of CSPAP. Our data provided evidence of validity and internal consistency for the final items included in subsequent analyses. The results of psychometric testing indicate that these measures can be used in future research investigating PE teachers’ socialization experiences and confidence with respect to being a PAL and involved with CSPAPs.

**Acculturation.** Based upon our examination of the relationships between socialization factors and RBSE, we found professional socialization and organizational socialization to be significant predictors of RBSE, whereas acculturation was not. Previous acculturation research found that pre-service and in-service teachers are highly influenced by their K-12 PE teachers and their positive or negative experiences within PE and sport [35,37,38]. Our results suggest that respondents felt their K-12 PE teachers were PALs and implemented PE programs. However, weaker associations were apparent for their PE teachers organizing expanded PA opportunities. This supports previous research indicating many PE teachers report positive K-12 school PE experiences [68] coupled with relatively few expanded PA opportunities [69,70,71]. However, previous acculturation literature suggests that lack of experiences with expanded PA opportunities as a K-12 student can motivate PE teachers to offer expanded PA experiences [71,72]. This was also the case for participants in the present study, although our findings indicated that acculturation was not a predictive factor of teachers’ confidence to serve as a PAL or be involved with a CSPAP. More research is needed to explore the association of PE teachers’ PA promotion experiences as a K-12 student and CSPAP-related RBSE.

**Professional socialization.** Regarding professional socialization, our results suggest the degree of PAL and CSPAP training PE teachers received in PETE varied. A plausible reason for this is the that the concept of a CSPAP is still relatively new, having emerged only in the last 15 years [12]. However, the demographic results are promising, as 20% of the respondents had first learned about CSPAP within their PETE programs. Furthermore, our results suggest that when PE teachers are trained during their PETE program to deliver quality PE and be PALs, they later feel confident to be a PAL and involved with CSPAP as in-service teachers. These results compliment the literature regarding the effectiveness of PAL [39] and CSPAP training [40,41,42,43] during professional socialization. In addition, previous RBSE research has suggested skill-specific training is an important intervention point to establish mastery and modeling to increase an individual’s RBSE. For example, training in interpersonal skills (e.g., networking, team building) and problem-solving skills (e.g., needs assessment, causal analysis) are relevant to enhance confidence to carry out a range of social, integrative, and proactive tasks [30], all of which are deemed necessary skills to be a PAL who is involved with a CSPAP [12,73]. Thus, we believe our findings support the need for PETE programs to provide PAL and CSPAP mastery learning experiences and CSPAP implementation modeling to further cultivate PE teachers’ RBSE for CSPAP implementation.

**Organizational socialization.** As with professional socialization, organizational socialization significantly predicted RBSE in this study. PE teachers felt confident to be a PAL and involved with a CSPAP, but organizational socialization factors (i.e., positive or negative support from administrators and school faculty) were perceived as consequential to their involvement. To overcome organizational socialization barriers, our findings and previous literature highlight the importance of PE teachers developing partnerships with school leaders [32,45,74,75] and fostering teacher and administrator support for effective CSPAP implementation and sustainability [49,50,76,77,78,79]. Furthermore, RBSE literature suggests supervisors (e.g., school administrators) who provide job enrichment (i.e., increase employee responsibility with decision making) enhance employee RBSE. The enriched work design means that employees have the discretion to take on broad roles and proactive tasks, which develop greater motivation to do so than in simplified job roles [30,80]. School administrators who advocate for PE teachers to be a PAL and implement a CSPAP may increase RBSE to take on these roles, although more research is needed.

**Barriers.** Participants in our study also viewed facilities/resources (e.g., available space, equipment, funding) as barriers to CSPAP implementation. Implementation and sustainability barriers may be due to (a) lack of financial support [78,81], (b) limited facilities available before/after school because of a focus on athletics [76,81], and (c) extracurricular obligations of PE teachers [81]. Based upon what is known in the organizational socialization and CSPAP literature, and as supported by the results of the current study, PE teachers are more confident to be a PAL and be involved with a CSPAP if they can collaborate and forge connections inside and outside the school community [76,78,81]. The RBSE literature supports these findings, as the work environment can be a key intervention point for enhancing RBSE. Organizations can offer opportunities for self-efficacy enhancing experiences such as developing “improvement groups” among employees who work to address problems within a work setting and build a stronger sense of self-efficacy to improve the workplace [30,82]. In the context of organizing and initiating a CSPAP, this would include collaborations between school, family, and community allies to develop and implement a CSPAP [6].

## 8. Strengths and Limitations

While this was one of the first investigations to examine the association of PE teachers’ CSPAP-related RBSE and all three phases of teacher socialization, this study has several limitations. First, the response rate for completing the survey was lower than in previous studies surveying public school faculty about CSPAPs (e.g., [44,47,49,50]). The low response rate limits the generalizability of the results and could have been different had the rate of survey responses been higher. However, survey respondents’ demographics were well distributed. Future survey studies with PE teachers should consider specific times of year to contact teachers for participation, which may elicit higher response rates. Another study limitation was the length of the survey (99 items), which may have discouraged participants from completing it. Furthermore, future research regarding the association of socialization and physical educator CSPAP involvement should employ additional qualitative research methods to allow for a deeper analysis of socialization factors, due to the contextual nature of teacher socialization. Additionally, future investigations warrant the investigation of demographic variables in explaining/differentiating PE teachers’ CSPAP-related RBSE. Demographics can be analyzed to provide further perspective on both shared and distinct perceptions of PE teachers in relation to socialization experiences and CSPAP-related RBSE. For instance, such research could be conducted using person-centered analyses (e.g., cluster analysis, latent profile analysis), as opposed to variable-centered analyses.

## 9. Conclusions

Available evidence suggests that preservice preparation for PE teachers’ expanded roles within the CSPAP framework is minimal [83,84]. Likewise, there appears to be a deficit in the implementation of CSPAPs nationally, with data from PE teachers [5] and school principals [49,50] indicating that approximately one third of schools in the U.S. do not have a CSPAP. The results of the present study indicate that, overall, PE teachers possess the confidence to take on the multiple roles involved with leading and implementing a CSPAP. However, higher levels of RBSE were apparent for teaching quality PE compared to serving as a PAL or being involved with other CSPAP components, which underscores the need to bolster support for PE teachers to engage in CSPAPs. Such support should include efforts to build PE teachers’ RBSE through both professional and organizational socialization experiences. Interventionists and teacher educators should consider incorporating professional training measures to develop PE teachers’ confidence to be a PAL and be implementors of CSPAPs. Additionally, cultivating PE teachers’ CSPAP-related confidence will require support from school leaders (i.e., administrators, faculty) and resources (i.e., equipment, compensation, facilities, faculty support).

## Figures and Tables

**Table 1 ijerph-19-12005-t001:** Participants’ self-reported demographics and school contexts.

		Participants (N = 259)
Highest level of education obtained	High school diploma/GED	2.4%
	Associates degree	4.1%
	Bachelors	41.2%
	Masters	27.6%
	Masters plus	21.2%
	Ph.D.	1.2%
	Ed.D.	2.4%
Employment Area Designation	Rural	40.6%
	Suburban	37.1%
	Urban	22.3%
Employment Region	West	41.5%
	South	17.2%
	Midwest	26.8%
	Northeast	14.5%
	0 Years	E * = 18.7%, M ** = 26.9%, H *** = 22.2%
Experience teaching physical education	1–5 Years	E = 40.6%, M = 29.7%, H = 40.7%
	6–10 Years	E = 16.1%, M = 17.9%, H = 10.4%
	11–15 Years	E = 7.1%, M = 14.5%, H = 11.1%
	16–20 Years	E = 9.0%, M = 3.4%, H = 8.1%
	21–25 Years	E = 3.9%, M = 4.1%, H = 3.7%
	26 or more	E = 4.5%, M = 3.4%, H = 3.7%
Total student enrolment	0–500	40.8%
	501–1000	33.1%
	1001–1500	15.4%
	1501–2000	7.7%
	2001–2500	2.4%
	2501+	0.6%
First learn about CSPAP	National conference	8.3%
	Regional conference	2.1%
	State conference	8.3%
	Website	6.3%
	Physical education teacher at your school	4.2%
	Physical education teacher not at your school	4.2%
	Classroom teacher at your school who is not a physical education teacher	0.7%
	Instructional coaches	0.7%
	Someone who holds a position in district-level leadership	4.2%
	Formal learning experiences in your pre-service teacher education program (e.g., PETE program)	20.1%
	Formal learning experiences in an in-service professional development workshop/training	5.6%
	Informal learning experiences (e.g., reading professional literature on your own)	2.1%
	This survey	31.9%
	National guidance documents	1.4%
Prior knowledge of CSPAP	Nothing	21.8%
	A little	24.7%
	Some	17.1%
	Fair amount	20.6%
	A lot	15.9%

* E = Elementary grades (i.e., K-5); ** M = Middle school grades (i.e., 6–8); *** H = High school grades (i.e., 9–12).

**Table 2 ijerph-19-12005-t002:** Descriptive statistics.

**Acculturation**	**Min**	**Max**	**M**	**SD**	**Skewness**	**Kurtosis**
As a K-12 student, at least one of my physical education teachers…						
Was considered the physical activity leader for the school (e.g., organized physical activity opportunities for students outside the classroom, promoted physical activity to staff).	1	6	4.48	1.514	–0.925	0.079
Implemented a physical education program that included: standards-based instruction, assessment of student learning, opportunities to learn, opportunities for moderate-to-vigorous physical activity.	1	6	4.56	1.428	–1.067	0.375
Organized physical activity opportunities for school staff/faculty (e.g., staff wellness programming, walking/jogging groups, staff training for physical activity promotion).	1	5	2.80	1.200	0.027	–0.569
Organized physical activity opportunities for my family/community (e.g., 5K events, family fitness nights at school, physical activity newsletters).	1	5	2.35	1.073	0.460	–0.228
Organized physical activity opportunities before/after school for all students (e.g., intramurals, physical activity clubs).	1	5	3.01	1.249	–0.097	–0.722
Organized physical activity opportunities during school for all students (e.g., classroom-based physical activity, structured recess, open-gyms).	1	5	3.10	1.240	–0.078	–0.782
**Professional Socialization**	**Min**	**Max**	**M**	**SD**	**Skewness**	**Kurtosis**
Based upon the survey definition of CSPAP training (i.e., Physical Education plus one or more components), my teacher certification program trained me to implement CSPAP as an in-service teacher.	1	6	3.91	1.346	–0.472	–0.223
My teacher certification program prepared me to develop…						
A physical education program that includes standards-based instruction, assessment of student learning, opportunities to learn, opportunities for moderate-to-vigorous physical activity.	1	6	5.09	0.877	–1.642	4.834
Additional physical activity opportunities before and/or after school (e.g., active transportation to school, intramurals, walk/run-a-thons, physical activity clubs, open gym).	1	6	4.11	1.162	–0.564	0.308
Physical activity initiatives during school (e.g., classroom-based physical activity, structured recess, physical activity assemblies, open gym).	1	6	4.27	1.088	–0.526	0.141
Physical activity initiatives involving family/community engagement (e.g., 5K events, family fitness nights at school, health fair).	1	6	3.94	1.209	–0.280	–0.059
Physical activity initiatives for school staff/faculty (e.g., fitness programs/events for teachers, health screening for teachers, staff training for physical activity promotion).	1	6	3.85	1.256	–0.129	–0.281
Establish partnerships with school/community stakeholders for physical activity initiatives (e.g., school administrators/faculty, universities, YMCAs, health department, parks and recreation, Boys/Girls Club).	1	6	4.08	1.213	–0.566	0.223
Evaluate current physical activity offerings in K-12 school environments (e.g., before/after school, during school, facilities, equipment resources).	1	6	4.26	1.190	–0.703	0.458
Develop joint use agreements for facility usage of physical activity initiatives.	1	6	3.80	1.295	–0.358	–0.273
Train school personnel on physical activity integration during school.	1	6	3.69	1.239	–0.132	–0.213
Market/promote physical activity initiatives.	1	6	4.13	1.200	–0.662	0.417
Implement CSPAP as a future in-service teacher.	1	6	3.84	1.370	–0.278	–0.528
**Organizational Socialization**	**Min**	**Max**	**M**	**SD**	**Skewness**	**Kurtosis**
My school promotes and/or supports active transport activities. (e.g., walking, cycling).	1	6	4.26	1.232	–0.835	0.702
Most teachers at my school provide activity breaks in the classroom, as a break, or as part of academic work.	1	6	4.15	1.269	–0.560	0.064
Most students in my school get more than one recess per day.	1	6	3.86	1.527	–0.447	–0.661
Community organized physical activity programs are available for all students on school grounds outside of the normal school day (e.g., YMCA/YWCA).	1	6	3.82	1.425	–0.408	–0.554
My school provides physical activity events for family and community members to participate.	1	6	3.69	1.317	–0.157	–0.355
My school provides physical activity classes/programs for faculty and/or staff. (e.g., walking/jogging, aerobics, yoga, basketball)	1	6	3.57	1.342	–0.142	–0.540
Indoor and outdoor physical activity facilities/resources (e.g., gym space, weight room, outdoor green space) positively influence my CSPAP involvement.	1	6	4.36	1.058	–0.998	1.671
Administrators expect me to implement CSPAP.	1	6	3.39	1.242	0.139	–0.426
Teachers/faculty expect me to implement CSPAP.	1	6	3.33	1.212	0.060	–0.178
Teachers/faculty positively influence my current CSPAP involvement.	1	6	3.63	1.131	–0.104	0.058
Family/community members expect me to implement CSPAP.	1	6	3.36	1.239	0.157	–0.397
Families/community positively influence my current CSPAP involvement.	1	6	3.55	1.114	–0.173	–0.028
Students positively influence my current CSPAP involvement.	1	6	4.13	1.035	–0.557	0.982
**Role Breadth Self-Efficacy (RBSE)**	**Min**	**Max**	**M**	**SD**	**Skewness**	**Kurtosis**
I feel confident implementing multiple components of CSPAP (e.g., before/after school physical activity, staff involvement).	1	6	4.89	1.057	–0.930	0.710
I feel confident implementing physical education program that includes standards-based instruction, assessment of student learning, opportunities for moderate-to-vigorous physical activity.	1	6	5.26	0.958	–1.575	2.871
I feel confident being a physical activity leader for my school(s) (e.g., organize physical activity opportunities for students outside the classroom, promote physical activity to staff and families/community).	1	6	4.94	1.030	–1.129	1.819

**Table 3 ijerph-19-12005-t003:** Model fit indices.

Model 1 (4 Factors)	Model 2 (3 Factors)	Model 3 (3 Factors, Simple Structure)	Model 4 (Structural Model)
1047.940	1215.710	939.252	1009.118
347	375	322	348
0.000	0.000	0.000	0.000
3.020	3.241	2.916	2.899
0.060	0.076	0.048	0.044
(0.054–0.066)	(0.070–0.082)	(0.042–0.054)	(0.038–0.050)
0.947	0.946	0.967	0.967
0.964	0.952	0.971	0.981
0.043	0.050	0.040	0.040

**Table 4 ijerph-19-12005-t004:** ESEM results.

**Acculturation**		**Estimate**	**SE**	**Estimate/SE**	**Two-Tailed** ***p* Value**
As a K-12 student, at least one of my physical education teachers…					
was considered the physical activity leader for the school (e.g., organized physical activity opportunities for students outside the classroom, promoted physical activity to staff.		0.722	0.087	8.270	0.000
implemented a physical education program that included: standards-based instruction, assessment of student learning, opportunities to learn, opportunities for moderate-to-vigorous physical activity).		0.538	0.074	7.307	0.000
organized physical activity opportunities before/after school for all students (e.g., intramurals, physical activity clubs).		0.527	0.075	7.035	0.000
organized physical activity opportunities for school staff/faculty (e.g., staff wellness programming, walking/jogging groups, staff training for physical activity promotion).		0.481	0.091	5.279	0.000
organized physical activity opportunities during school for all students (e.g., classroom-based physical activity, structured recess, open-gyms).		0.405	0.081	5.020	0.000
Please tell us more about your CSPAP-related participation experiences as a K-12 student.		“I didn’t have much participation, so it motivates me to give my students more opportunities” (Henry)			
**Professional Socialization**		**Estimate**	**SE**	**Estimate/SE**	**Two-Tailed** ***p* Value**
My teacher certification program prepared me to…					
develop physical activity initiatives for school staff/faculty (e.g., fitness programs/events for teachers, health screening for teachers, staff training for physical activity promotion).		0.887	0.051	17.310	0.000
evaluate current physical activity offerings in K-12 school environments (e.g., before/after school, during school, facilities, equipment resources).		0.885	0.052	16.950	0.000
develop additional physical activity opportunities before and/or after school (e.g., active transportation to school, intramurals, walk/run-a-thons, physical activity clubs, open gym).		0.877	0.056	15.703	0.000
establish partnerships with school/community stakeholders for physical activity initiatives (e.g., school administrators/faculty, universities, YMCAs, health department, parks and recreation, Boys and Girls Club).		0.866	0.052	16.549	0.000
market/promote physical activity initiatives.		0.862	0.055	15.537	0.000
develop physical activity initiatives involving family/community engagement (e.g., 5K events, family fitness nights at school, health fair).		0.850	0.052	16.510	0.000
train school personnel on physical activity integration during school.		0.832	0.053	15.576	0.000
implement CSPAP as a future in-service teacher.		0.806	0.053	15.281	0.000
Based upon the survey definition of CSPAP (i.e., Physical Education plus one or more components), my teacher certification program trained me to implement CSPAP as an in-service teacher.		0.736	0.058	12.804	0.000
My teacher certification program prepared me to…					
develop physical activity initiatives during school (e.g., classroom-based physical activity, structured recess).		0.735	0.058	12.624	0.000
develop joint use agreements for facility usage of physical activity initiatives.		0.728	0.055	13.232	0.000
develop a physical education program that includes: standards-based instruction, assessment of student learning, opportunities to learn, opportunities for moderate-to-vigorous physical activity.		0.516	0.070	7.329	0.000
Please tell us more about your training experiences with PA promotion (e.g., CSPAP).		“In my certification program I created a staff wellness program, and I wrote an article for SHAPE dissecting aspects of community relations & resources to be used as part of a CSPAP (Sarah)			
**Organizational Socialization**		**Estimate**	**SE**	**Estimate/SE**	**Two-Tailed** ***p* Value**
Teachers/faculty expect me to implement CSPAP.		0.872	0.060	14.557	0.000
Teachers/faculty positively influence my current CSPAP involvement.		0.815	0.058	14.021	0.000
Administrators expect me to implement CSPAP.		0.775	0.059	13.237	0.000
Family/community members expect me to implement CSPAP.		0.742	0.061	12.167	0.000
Families/community positively influence my current CSPAP involvement.		0.720	0.060	12.067	0.000
My school provides physical activity events for family and community members to participate.		0.669	0.062	10.709	0.000
Students positively influence my CSPAP involvement.		0.652	0.065	10.059	0.000
Most students in my school get more than one recess per day.		0.471	0.069	6.781	0.000
My school provides physical activity classes/programs for faculty and/or staff. (e.g., walking/jogging, aerobics, yoga, basketball).		0.464	0.068	6.826	0.000
My school promotes and/or supports active transport activities. (e.g., walking, cycling).		0.450	0.067	6.669	0.000
Indoor and outdoor physical activity facilities/resources (e.g., gym space, weight room, outdoor green space) positively influence my CSPAP involvement.		0.404	0.068	5.929	0.000
Community organized physical activity programs are available for all students on school grounds outside of the normal school day (e.g., YMCA/YWCA).		0.362	0.068	5.310	0.000
Please tell us about how where you teach influences your current CSPAP involvement.		“I have great support from administration and teachers to implement CSPAP. (Elaine)			
**Role Breadth Self-Efficacy (RBSE)**		**Estimate**	**SE**	**Estimate/SE**	**Two-Tailed** ***p* Value**
I feel confident implementing multiple components of CSPAP (e.g., before/after school physical activity, staff involvement).		0.920	0.021	44.331	0.000
I feel confident implementing physical education program that includes standards-based instruction, assessment of student learning, opportunities for moderate-to-vigorous physical activity.		0.870	0.026	33.955	0.000
I feel confident being a physical activity leader for my school(s) (e.g., organize physical activity opportunities for students outside the classroom, promote physical activity to staff and families/community).		0.756	0.027	27.569	0.000
Please tell us more about the factors that influence your beliefs and confidence to implement CSPAP.		“Confidence isn’t a problem; I only get paid to teach physical education and health during school hours” (Lois)			
**Path Coefficients**		**Estimate**	**SE**	**Estimate/SE**	**Two-Tailed** ***p* Value**
	Acculturation→RBSE	–0.005	0.065	–0.081	0.935
	Professional Socialization→RBSE	0.246	0.058	4.246	0.000
	Organizational Socialization→RBSE	0.353	0.061	5.839	0.000
**Covariances**		**Estimate**	**SE**	**Estimate/SE**	**Two-Tailed** ***p* Value**
	Professional Socialization—Acculturation	0.119	0.115	1.040	0.298
	Organizational Socialization—Acculturation	0.142	0.108	1.312	0.190
	Organizational—Professional Socialization	0.470	0.058	8.039	<0.001
**Correlations**		**Estimate**	**SE**	**Estimate/SE**	**Two-Tailed** ***p* Value**
	Professional Socialization—Acculturation	0.147	0.121	1.215	0.225
	Organizational Socialization—Acculturation	0.187	0.120	1.558	0.120
	Organizational—Professional Socialization	0.497	0.093	5.344	<0.001

## Data Availability

All research activities were approved by the first author’s university Institutional Review Board prior to the initiation of this study.

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
