# Peer review of "Association of Physical Educators’ Socialization Experiences and Confidence with Respect to Comprehensive School Physical Activity Program Implementation"

_ijerph, 2022, doi:10.3390/ijerph191912005_

Round 1
Reviewer 1 Report
Manuscript ID: ijerph-1896051
Title: Association of Physical Educators’ Socialization Experiences and Confidence with Respect to Comprehensive School Physical Activity Program Implementation
Overall, I am very impressed with this paper. It was a joy to review this as it flowed so well, and it really (to me) seemed like an already published manuscript. It is in great shape, and I especially commend the authors for great citing of relevant and recent research, and the writing is easy to follow even when you are explaining complex topics.
The theoretical framework is explained thoroughly and is easy to follow. Great job.
I would say the one issue I found in section 2: “Further, efficacious PE teachers are, and open to, new ideas and roles (Siedentop, 89 2002; Tsangaridou, 2002).” This seems like it needs a re-write.
The one thing I cannot comment on as well is the use of exploratory structural equation modeling and the (to me) complex analysis. To me, on the surface, it seems correct but I cannot judge completely.
One point was surprising… 32% learned about CSPAP through THIS survey!! That is pretty interesting (not that you need to respond to this, but it’s fascinating that 32% of the respondents completed a 99 question survey and didn’t even know CSPAP).
Everything in this paper just flows so well. I am really impressed. I am literally writing more now since this review is so short! I truly have very little to add for this paper to get published! This is definitely in the top 5% of first round reviews I’ve done. Great work!
Author Response
Thank you for all your feedback and helping us improve the manuscript.
Please see the attachment for responses to reviewer comments.

Reviewer 2 Report
The paper is well written and logically structured. Introduction is clear and provides a rationale for the study. Methodological approaches are very well explained and establish a clear pathway to the results section. Consider the placement of Table 1 (participant demographics): two issues to consider, 1. is this table needed? could the information in the table be condensed into a few lines, 2. If including, should it not be at the start of the results section?
Considering the emphasis in the methodology relating to the demographic spread of participants, I thought that the analysis may have drawn on this information i.e. differences in response to particular items relating to years of teaching experience, school level they taught, how does employment area designation impact prevalence or not of CSPAP in schools?
The question is original and well defined and the results provide an advancement of the current knowledge.
Conclusion can be improved - rework conclusion so the main headlines from the findings are clear and more obvious for the reader.
Specifics below in relation to particular lines, but otherwise it is very clearly written.
Line 89 - 'Further, efficacious PE teachers are, and open to, new ideas and roles' (Siedentop, 89 2002; Tsangaridou, 2002). This sentence needs fixing - not clear.
Lines 86 - 102: consider the currency of the literature and aim for inclusion of more up to date evidence.
Lines 154-156: 'then trainings during these phases of teachers’ career development 154 could include learning experiences designed to specifically build PE teachers’ confidence 155 in line with RBSE'. Change 'trainings' - best to use education
Author Response
Thank you for your feedback and helping us improve the manuscript.
Please see the attachment for responses to reviewer comments.

Reviewer 3 Report
Well-written study and competent text, it is a pleasure to read it. However a few questions have arisen.
Q1: How about rotation? What rotation method you used in ESEM? I think it should be mentioned in the text.
Q2: What are correlations between latent factors?
Q3: Mplus 8 uses FIML automatically for missing values conditions of MAR so it seems that lines 364-367 are meaningless or is there deeper purpose for doing extra imputations? However you can do that way too if it seems to be better way.
Author Response

(The authors gave the same response as above.)
